# Variation in Lipid Species Profiles among Leukemic Cells Significantly Impacts Their Sensitivity to the Drug Targeting of Lipid Metabolism and the Prognosis of AML Patients

**DOI:** 10.3390/ijms24065988

**Published:** 2023-03-22

**Authors:** Caroline Lo Presti, Yoshiki Yamaryo-Botté, Julie Mondet, Sylvie Berthier, Denisa Nutiu, Cyrille Botté, Pascal Mossuz

**Affiliations:** 1Team “Epigenetic and Cellular Signaling”, Institute for Advanced Biosciences, University Grenoble Alpes (UGA), INSERM U1209/CNRS 5309, 38700 Grenoble, France; clopresti@chu-grenoble.fr (C.L.P.); jmondet@chu-grenoble.fr (J.M.); nutiu.denisa@gmail.com (D.N.); 2Department of Biological Hematology, Institute of Biology and Pathology, Hospital of Grenoble Alpes (CHUGA), CS 20217, 38043 Grenoble, CEDEX 9, France; 3Team “Apicolipid”, Institute for Advanced Biosciences, University Grenoble Alpes (UGA), INSERM U1209/CNRS 5309, 38700 Grenoble, France; yoshiki.botte-yamaryo@univ-grenoble-alpes.fr (Y.Y.-B.); cyrille.botte@univ-grenoble-alpes.fr (C.B.); 4Department of Molecular Pathology, Institute of Biology and Pathology, Hospital of Grenoble Alpes (CHUGA), CS 20217, 38043 Grenoble, CEDEX 9, France; 5Platform of Cytometry, Institute of Biology and Pathology, Hospital of Grenoble Alpes (CHUGA), CS 20217, 38043 Grenoble, CEDEX 9, France; sberthier@chu-grenoble.fr

**Keywords:** acute myeloid leukemia, metabolism, lipid species, inhibition of FAO

## Abstract

Several studies have linked bad prognoses of acute myeloid leukemia (AML) to the ability of leukemic cells to reprogram their metabolism and, in particular, their lipid metabolism. In this context, we performed “in-depth” characterization of fatty acids (FAs) and lipid species in leukemic cell lines and in plasma from AML patients. We firstly showed that leukemic cell lines harbored significant differences in their lipid profiles at steady state, and that under nutrient stress, they developed common mechanisms of protection that led to variation in the same lipid species; this highlights that the remodeling of lipid species is a major and shared mechanism of adaptation to stress in leukemic cells. We also showed that sensitivity to etomoxir, which blocks fatty acid oxidation (FAO), was dependent on the initial lipid profile of cell lines, suggesting that only a particular “lipidic phenotype” is sensitive to the drug targeting of FAO. We then showed that the lipid profiles of plasma samples from AML patients were significantly correlated with the prognosis of patients. In particular, we highlighted the impact of phosphocholine and phosphatidyl-choline metabolism on patients’ survival. In conclusion, our data show that balance between lipid species is a phenotypic marker of the diversity of leukemic cells that significantly influences their proliferation and resistance to stress, and thereby, the prognosis of AML patients.

## 1. Introduction

AML comprises a group of malignant neoplastic disorders characterized by accumulation in the blood and/or in the bone marrow of immature white blood cells called leukemic cells, together with a deficiency in normal hematopoiesis. If conventional treatment based on intensive chemotherapy allows complete remission for 60 to 85% of patients, the prognosis of AML remains very negative, with a global survival rate of around 25 to 30%. In recent years, several studies linked this aggressiveness and the bad prognosis of AML to the ability of the leukemic cells to reprogram their metabolism in order to satisfy their high metabolic needs and to resist chemotherapy. In addition to the universally known Warburg effect, which involves a shift from oxidative phosphorylation as the main energy-producing pathway to high-rate aerobic glycolysis coupled with lactic acid fermentation in the cytosol, many metabolic pathways are deregulated in leukemic cells, sometimes in direct correlation with genetic alterations [1]; this suggests multimodal metabolism in leukemogenesis. Among these metabolic pathways, lipid metabolism plays a particular role that has already been described in many solid tumors. Alterations, such as accentuated FAO and de novo lipid synthesis, could promote cancer cell survival and contribute to resistance to chemotherapeutic agents and metastasis [2] through an increase in energy production and in lipid species necessary for the production of new cells. The overexpression of fatty acid synthase (FASN) gives cancer cells a proliferative advantage; conversely, lipolysis, through the activation of lipoprotein lipase (LPL), could increase the availability of FAs to levels necessary for cancer cell proliferation.

In AML, a recent study showed that patients with rapidly progressive AML who did not respond to treatment had alterations in their lipid metabolism [3]. Interestingly, both aspects of lipid metabolism were implicated. On one hand, the degradation of FA through FAO may contribute to chemotherapy resistance. Indeed, resistant AML cells show overexpression of CD36, an FA transporter, at their membranes, and very intense FAO. This particular role of FAO in the survival of leukemic cells was also underlined in a demonstration of the overexpression of the enzyme 3-very long chain acyl-CoA dehydrogenase (VLCAD) in AML, of which a decrease in proliferation can be induced by gene attenuation or pharmacological inhibition via alteration of the Krebs cycle [4]. Additionally, a recent study demonstrated that glycerophospholipid metabolism is involved in drug resistance in chronic myeloid leukemia through downregulation of the G0/G1 switch gene 2 (G0S2), which confers worse overall survival to patients [5]. Similarly, Carnitine Palmitoyltransferase 1A (CPT1A), a key enzyme in carnitine-dependent transport across the mitochondrial inner membrane of FAs, was found to be overexpressed in bone marrow samples from AML patients with poor prognosis compared to those with normal bone marrow [6]. These data provide a rationale for studying the impact of etomoxir (Eto), which irreversibly inhibits CPT1A, thereby preventing the entrance of fatty acids into the mitochondria, and their beta-oxidation.

On the other hand, the synthesis of FAs also plays a critical role. It was shown that low expression of ATP Citrate Lyase (ACL), the enzyme that initiates lipid synthesis through Acetyl-CoA production, is associated with favorable prognosis in patients with AML, and that discontinuation of cell growth was achieved when ACL was inhibited in leukemia cell models [7]. Similarly, an anti-leukemic role of Acetyl-CoA Carboxylase 1 (ACC1), the enzyme that catalyzes the production of Malonyl-CoA from acetyl-CoA, has been shown [8].

In AML patients, several lipid patterns representative of different subtypes of AML have recently been identified via mass spectrometry. Indeed, significant differences concerning modulation of the synthesis of ceramides and sphingolipids have been highlighted in patients presenting with t(8;21) compared to those presenting with an inv(16) or a normal karyotype [9]. Additionally, Stuani et al. compared the dysregulation of lipid metabolism in a cellular model of AML mutated for the IDH1 gene to that in a non-mutated model [10]. These differences are mainly characterized by an increase in phosphatidylinositol, sphingolipids, free cholesterol and monounsaturated fatty acids in IDH-mutated cells. This, again, highlights the impact of the cytogenetic characteristics and mutational status of AML patients on cellular metabolism. Finally, a recent study showed a decrease in total FAs in the plasma of patients with AML, as well as the attenuation of plasmatic phosphocholine, triglycerides and cholesterol esters [11]. In addition, arachidonic acid appears to be increased in AML plasma samples, suggesting its involvement in the cancerous phenotype [12].

Interestingly, lipids may play a role in the expression and functionality of certain oncogenes. It was shown that the aberrant localization of FLT3-ITD at the level of the endoplasmic reticulum depended on the palmitoylation of the receptor, and that the pharmacological inhibition of the depalmitoylation of the receptor had a synergistic effect on the inhibitor of FLT3 [13]. Similarly, in an MLL-AF9 mouse model, a high-fat diet was shown to increase FLT3-dependent signaling via clusterization of the receptor in lipid rafts, and thus, promote the emergence of leukemic clones [14]. Picou et al. tested the impact of polyunsaturated FAs (PUFAs) on acute myeloid leukemia cell lines and on fresh leukemic cells from patients, and induced the inhibition of mitochondrial respiration, and increased glycolysis and oxidative stress, which lead to the death of leukemic cells [15]. However, Gyan et al. did not succeed in demonstrating the significant impact of FAs from fish oil emulsion on the complete remission of AML patients [16].

Altogether, these data suggested that the reprogramming of lipid metabolism and the balance of lipid species could play a role in improving the prognosis of AML. In this context, we performed extensive characterization of the lipidome of human leukemic cell lines and plasma samples from AML patients, to decipher the characteristics of the lipid species and evaluate those of potential interest for targeting lipid metabolism in AML.

## 2. Results

### 2.1. Characterization of Lipid Profiles of Different Leukemic Cell Lines

#### 2.1.1. Quantification and Characterization of Total Fatty Acids

The overall quantification of total fatty acids (glycerolipids and free fatty acids) in five leukemic cell lines cultured under normal conditions shows significant differences between the cell lines (Figure 1A). In particular, the K562 cell line exhibits a significant increase in total FAs compared to the other four cell lines (*p* < 0.0001), with an average of 934.1 nmol/10^7^ cells. HEL contains an average of 423.3 nmol/10^7^ cells, followed by OCI-AML3 cells, with an average of 302.2 nmol/10^7^ cells, and then, KG1, with 234.9 nmol/10^7^ cells. HL60 is the cell line with the lowest amount of total fatty acids (136 nmol/10^7^ cells).

To obtain detailed information on the composition of the different fatty acids, we chose to compare the two cell lines K562 and KG1, which represent cell lines with very high and low amounts of total FAs, respectively. Interestingly, both cell lines harbored different profiles with overexpression of different fatty acids: C16:0 (palmitic acid), and cis and trans C18:1; C20:1 (gadoleic acid) and C20:3 (DGLA or dihomo-gamma-linolenic acid) were overexpressed in K562 compared to KG1, and conversely, C16:1 (palmitoleic acid), C18:0 (stearic acid) and C20:5 (eicosapentaenoic acid) were overexpressed in KG1 compared to K562 (Figure 1B). The concentrations of other FAs, C14:0 (myristic acid), C18:2 (linoleic acid), C20:4 (arachidonic acid) and C22:6 (docosahexaenoic acid or DHA), were quite similar.

#### 2.1.2. Characterization of Lipid Species

In order to improve the characterization of the lipid profiles of the two cell lines, we carried out a comparative study of the lipid species. In the basal state, many lipid species (>10) were expressed by the two cell lines. However, the lipid species composition of the KG1 and K562 cell lines showed significant differences. The main differences observed were a significant increase in phosphatidyl-choline ((P)C) (*p* < 0.05), phosphatidyl-ethanolamine ((P)E) (*p* < 0.01), a triacylglycerol/cholesteryl ester mixture (TAG/CE) (*p* < 0.05) and cardiolipin (CL) (*p* < 0.001) in K562 compared to KG1. Interestingly, we did not observe (as we did for FAs) that some lipid species were overexpressed in KG1 compared to K562 (Figure 1C).

### 2.2. Impact of Starvation on Lipid Metabolism in Leukemic Cell Lines

Since lipid catabolism is a main pathway of energy production, especially in nutritional stress, we studied variations in lipid profiles (total fatty acids and lipid species) after starvation (absence of FBS) in the two cell lines. Under these conditions, we initially observed a decreased rate of proliferation with a decrease in the percentage of viable cells at 24 and 48 h, but the cells quickly adapted and continued to proliferate, with levels rising to reach viability close to that under normal conditions.

#### 2.2.1. Impact on Total Fatty Acids

Under starvation conditions, the total amount of FAs is significantly reduced in both cell lines compared to their basal state (Figure 2A,B). Regarding FA composition, very similar modifications in the profiles of the two cell lines were observed (Figure 2C,D), characterized by a reduction in the quantity of stearic acid (C18:0) (*p* < 0.001), as well as PUFAs, DGLA (C20:3), arachidonic acid (C20:4), EPA or eicosapentaenoic acid (C20:5) and DHA (C22:6) (*p* < 0.001) in both cell lines. Interestingly, we observed similar results in the two cell lines, with a very significant increase in the amount of palmitoleic acid (C16:1) in starvation compared to the basal state (*p* < 0.001).

#### 2.2.2. Impact on Lipid Species

As observed with fatty acids, the two leukemic cell lines under starvation conditions show similar decreases in several lipid species. A significant decrease in (P)C (*p* < 0.05) and cardiolipin (*p* < 0.001) was thus observed in both KG1 and K562. Other species were reduced in the two cell lines, but with their amplitude depending on the cell line; these were the TAG/CE mixture and lyso-biphosphatidic acid (LBPA), which were significantly reduced only in K562 (*p* < 0.05). Conversely, phosphatidyl-serine ((P)S), sphingomyelin (SM) and a particular lipid species, PE-Ceramide, which was found only in KG1, were significantly reduced in KG1 (*p* < 0.01) but remained stable in K562 (Figure 2E,F).

### 2.3. Impact of FAO Inhibition on the Viability of AML Cell Lines KG1 and K562

#### 2.3.1. Impact of Etomoxir on Cell Survival

Our results suggest that the modulation of lipid species is one of the mechanisms used by leukemic cells to counteract starvation. In this context, we studied the impact of etomoxir (Eto), an inhibitor of FAO, on the survival of leukemia cells. To determine the sensitivity of each cell line, we first assessed the effect of Eto on the viability of the two AML cell lines KG1 and K562, cultivated in normal medium culture with FBS. For this first set of experiments, we evaluated, based on data from the literature, two concentrations of Eto: 200 and 400 μM, respectively. As shown in Figure 3A,B, the percentage of viable cells decreased significantly in both cell lines with dose and time, demonstrating a dose- and time-dependent inhibitory effect of Eto on these leukemic cells. However, the sensitivity levels of the two cell lines were significantly different. The percentage of viable K562 cells after 24 h and 48 h of treatment with Eto was significantly more decreased compared to that of KG1 cells, showing that K562 was more sensitive to the action of Eto than KG1. The calculated IC_50_ values of etomoxir were 113.7 µM and 320 µM for K562 and KG1, respectively, confirming their differences in sensitivity.

#### 2.3.2. Impact of Starvation on Sensitivity to Etomoxir

Secondly, we decided to evaluate whether the combination of Eto with starvation could significantly improve leukemic cell death. Taking into account the differences in sensitivity observed under normal conditions, we used different concentrations of Eto, between K562 and KG1 (200 and 400 µM, respectively) for this evaluation in the absence of serum.

As shown in Figure 3C,D, the viability of cells decreased as expected when they were deprived of nutrients, but did not reach high levels (greater than 50%) of dead cells, suggesting that both cell lines could counteract FBS deprivation. Of note, we observed that KG1, despite its lower reserve of total fatty acids, was not more sensitive to FBS withdrawal compared to K562. When the cells were cultivated with etomoxir in serum-deprived medium, we observed a dramatic increase in cell death, reaching almost complete mortality after 48 h. Interestingly, at 24 h, the KG1 cells were almost all dead compared to the K562 cells, of which around 20% were still alive.

### 2.4. Impact of FAO Inhibition on Total Lipid Abundance and Lipid Profiles in KG1 and K562

As the culture with Eto suggested that FAO is critical for leukemic cell survival, we decided to analyze the impact of FAO inhibition on total lipid abundance and lipid profiles in KG1 and K562, in correlation with their sensitivity to Eto. Based on the results on viability in the presence of Eto, and in order to decipher how the balance of different lipid species could modulate sensitivity to Eto, we analyzed the impact of a low dose of Eto (100 µM). Indeed, at this dose, K562 (IC_50_ = 113 µM) was sensitive to Eto, and conversely, KG1 (IC_50_ = 320 µM) was resistant to Eto. Firstly, we did not observe a significant quantitative modification of the total amount of FAs after incubation with Eto in both cell lines, but we observed qualitative variation between the two cell lines. Indeed, we observed that KG1 harbored more alteration (mainly a decrease) in FA levels than K562. We observed a very significant decrease (*p* < 0.001) in palmitic acid (C16:0), stearic acid (C18:0) and oleic acid (C18:1) in KG1 (Figure 4A). The only shared variation was for arachidonic acid (C20:4), which decrease significantly in KG1 and K562 (Figure 4A,B). Moreover, the relative proportion of eicosenoic acid (C20:1) increased significantly in KG1, whereas we did not observe significant variation in K562 for this fatty acid. Regarding lipid species, the main result was a clear decrease in TAG (*p* < 0.001), associated with a weak but significant decrease in (P)C (*p* < 0.05), which was only observed in KG1 cells (Figure 4C,D).

### 2.5. Impact of FAO Inhibition on Mitochondrial Respiration

To evaluate the impact of FAO inhibition on mitochondrial function, using the SeaHorse XF method, we evaluated the respiratory profiles of KG1 and K562, without Eto (control) and with increasing concentrations of Eto, at 24 h of culture (Figure 5A,B). Based on the results of viable cells, we decided to firstly use 200 µM of Eto since this concentration impacted the cell viability of both cell lines without inducing too many dead cells that could bias the measure of respiratory profiles. We also analyzed, as for the lipid profiles, the impact of a lower dose of Eto (100 µM), which could yield interesting data between K562 (considered sensitive to this dose) and KG1 (considered resistant to this dose).

Firstly, we observed a significant decrease in basal respiration (BR) in KG1 and K562 cultured with Eto, when compared to the control cultures (Figure 5C). The impact started at 100 µM in both cell lines and increased significantly at 200 µM. Similarly, we observed a decrease in the maximal respiration (MR) proportional to the concentration of etomoxir (Figure 5D). ATP production followed the same trend as BR and MR and was decreased in the case of cultures with etomoxir, dose-dependently, but this decrease was not significant. The spare respiratory capacity (SRC), which is an indicator of the capacity of cells to respond to increased energy demand, remained stable, and even increased slightly in K562 compared to KG1 (not significant).

### 2.6. Characterization of Lipid Profiles in AML Patients: Correlation with Their Prognosis

First, we analyzed the composition of total fatty acids in the plasma of patients with a favorable (n = 21) versus unfavorable (n = 18) prognosis (according to ELN 2017 [17]). According to the statistical analysis, no significant difference was observed between the two groups of patients concerning the total content of FAs (Figure 6A), as well as the total content of phospholipids (PLs) and neutral lipids. We did not observe any major differences between patients with a favorable and unfavorable prognoses (respectively, *p* = 0.28 and *p* = 0.20) (Figure 6B,C).

To further our investigation, we compared detailed information on the composition of the lipid species of five patients representative of each group. Concerning FA composition, even though we observed some trends, we did not identify significant differences associated with the prognosis of patients (Figure 6D). Concerning phospholipids, we were able to identify eight compounds in the ten patients. On the one hand, we identified sphingomyelin, phosphatidyl-choline, phosphatidyl-inositol ((P)I) and phosphatidyl-ethanolamine, which correspond to phospholipids (PLs); on the other hand, we identified a mixture of diacylglycerol and cholesterol (DAG-Cho), free fatty acids (FFAs), triacylglycerol (TAG) and cholesteryl esters (CEs), which correspond to neutral lipids (NLs). After analyzing each lipid spot, we were able to demonstrate a significant increase in SM, (P)C (*p* < 0.05), (P)I (*p* < 0.01) and FFAs (*p* < 0.01) in the plasma of patients with an unfavorable prognosis (Figure 6E).

## 3. Discussion

In this study, we firstly aimed to decipher the characteristics of lipid species in leukemic cells that could be of interest for targeting lipid metabolism for AML survival. Our first results show that, in their basic state, the five leukemic cell lines, which were representative of a certain diversity of AML, were characterized by very different total quantities of fatty acids, with a ratio of 1 to 7 between the K562 (the highest) and HL60 (the lowest) cell lines. To further characterize the relative proportion of fatty acids and lipid species, independently of the initial quantity of fatty acids, we chose to analyze K562 and KG1, which presented very different levels of FAs at baseline, suggesting very different lipids compositions. In fact, the qualitative analysis of their lipid contents showed very clear differences in the relative proportion of lipid species, demonstrating the diversity of the lipidome of leukemic cells. These results show that, in accordance with the results that we have previously published on the HRMAS profiles of these cell lines [18], there are clear differences in the expression levels of metabolites between leukemic cell lines. The most probable hypothesis is a direct link between the metabolic profile and the genetic status of leukemic cells, as has already been shown in patients with t(8;21), inv(16) or IDH mutation [9,10].

These differences in the pattern of lipid expression suggest that leukemic cells might develop different lipid metabolism reprogramming under stress. As our goal was to study the adaptation of leukemic cells to global metabolic stress, and not to a particular deficiency, we chose, in our experiments, to use global serum deprivation. Although the use of serum-free medium that was selectively supplemented with a particular type of lipid species would be more precise, it would not have provided an overall picture of lipid reprogramming in this situation. Moreover, FBS is the major source of lipids in the medium, and the removal of serum is now valid and has been used in a number of publications; moreover, to date, there is no controlled medium whose exact and necessary FA composition is known.

Under these conditions, we confirmed our previous results, which showed the rapid adaptation of these two cell lines to starvation, allowing them to survive thanks to metabolic reprogramming [18]. Regarding the lipid level, the first indicator of this adaptation was a significant decrease in their overall level of fatty acids after 48 h of serum deficiency, suggesting that the “basic quantitative reserves” of lipids in the cells are, in themselves, a protective mechanism. A decrease in lipid storage, such as DAG/cholesterol and TAG/cholesteryl ester, can also be attributed to the mobilization of lipids for survival. Interestingly, we observed significant variation in the lipid species that showed many similarities between the two cell lines, suggesting that under this stress condition, the leukemic cells set up common pathways, independently of the differences observed in their basal state. In particular, we observed a decrease in phosphatidyl serine, phosphatidyl-ethanolamine and phosphatidyl-choline, which suggests that leukemic cells use phosphatidyl-choline and its substrates to survive. The deregulation of phosphatidyl-choline metabolism has been described in many cancers [19]. Several papers have noted changes in phosphatidyl-choline levels in cancers, particularly those of the prostate, ovaries and lung [20,21]. Similarly, the dysregulation of phosphatidyl-ethanolamine and phosphatidyl-inositol has been reported in prostate cancer. Moreover, Zheng et al. showed that under serum deprivation, human breast cancer cells exhibited increased phospholipase D (PLD) activity, leading to a decrease in (P)C (via the activation of its catabolism) and increased migration and metastasis potential [22]. We also observed a significant decrease in arachidonic acid, high levels of which have been described in AML patients with a poor prognosis [11]. Hence, our results show, for the first time, and to our knowledge, that the remodeling of lipid species is a major and shared mechanism of adaptation to stress in leukemic cells. This role was highlighted in the mechanism of sensitivity of acute promyelocytic leukemia (APL) to all-trans-retinoic acid (ATRA). Highly unsaturated fatty acids (HUFA) containing phosphatidyl-ethanolamine (HUFA-(P)E), such as those found in salmon testes, significantly enhanced dbcAMP-induced cell differentiation [23]. This effect was suppressed by a protein kinase C inhibitor, suggesting that HUFA-(P)E might enhance dbcAMP-induced differentiation through the modulation of the protein kinase C signaling pathway in HL60 cells.

Overall, these results suggest that therapeutic targeting of lipid metabolism could be conceived as a complementary treatment to increase the therapeutic efficacy of induction chemotherapy. To assess this hypothesis, we studied the impacts of treating leukemic cells with etomoxir (Eto), which irreversibly inhibits CPT1A, thereby preventing the entrance of fatty acids into the mitochondria, and their beta-oxidation. In the presence of Eto, we observed a decrease in viability time and dose dependence in both cell lines, which increased dramatically in the absence of serum. However, sensitivity to Eto was significantly different between the two cell lines. The seahorse analysis of mitochondrial respiration demonstrated that in both cell lines, Eto—by inhibiting FAO—definitively reduces ATP production, even though KG1 was less impacted than K562, which was consistent with what we observed regarding viability.

The fact that K562 was significantly more sensitive compared to KG1, despite higher levels of FAs at steady state, could suggest deeper dependence of K562 cells on lipid metabolism, which is prevented by higher levels of lipid stock. Moreover, the IC_50_ level in KG1 was more than 300 µM, and we cannot exclude that at these doses, as well as off-target effects such as the inhibition of Complex I of the electron transport chain (ETC) and the production of reactive species of oxygen (ROS), are the main factors responsible for cell death, as previously described [24].

In addition, comparison of the modifications of lipid profiles induced by starvation or inhibition due to FAO, respectively, shows that if the two leukemia cell lines, despite their genotypic and phenotypic differences, have developed very closed mechanisms to fight against starvation, their levels of sensitivity to metabolic inhibitors are different and would depend on their initial lipid profiles. On one hand, K562 cells, with a dose close to their IC_50_ value, present a blockade of the use of most of their fatty acids (in coherence with the mechanism of action of etomoxir), since their rates and relative proportions show very few differences compared to their basal profiles. In addition, no modification of the lipid species was observed in this cell line, suggesting that the blockade of FAO actually induced the death of K562 cells. Conversely, the KG1 cells, which are resistant to a dose of 100 µM of Eto (IC_50_ = 320 µM), showed very significant variations (*p* < 0.001) in the rates and relative proportions of several fatty acids. This decrease in the presence of Eto, which blocks FAO, together with the resistance of KG1 cells to etomoxir, suggests that KG1 cells have developed a mechanism of resistance against Eto that involves certain FAs and/or storage lipids. We can hypothesize that these lipid species are used for the renewal of new leukemic cells through their structural role, but also that they could be implicated in mechanism of drug resistance. Indeed, glycosphingolipid biosynthesis was shown to regulate drug sensitivity in clinically relevant models of drug-resistant chronic myeloid leukemias, through modulation of both the expression and activity of the drug efflux proteins ABCB1 and ABCC1 [25].

Altogether, the results of this first part show that in AML, one can discriminate leukemic cells that are highly dependent on FAO, for which the use of an inhibitor of FAO should be efficient, and conversely, leukemic cells that are independent of FAO, for which an inhibitor of FAO would not be a reliable additional therapy. Therefore, the targeting of lipid metabolism should be proposed only for a subset of AML patients who display a sensitive “lipidic phenotype”.

Leukemic cell lines represent a questionable model of what happens in patients. This is why we reinforced the in vitro analysis with an ex vivo analysis of patient samples, in order to assess the role of the patients’ lipidome on their survival. We firstly compared total fatty acids, phospholipids and neutral lipids among a cohort of 39 AML patients with high (n = 21) or low (n = 18) prognoses (according to ELN 2017 [17]). In addition, as all investigations of lipid species (FA and phospholipid composition) using LC-MS and GC-MS are very time-consuming experiences, require very high-level expertise and could not be carried out on large series of patients, we further selected five patients belonging to the high or low prognostic subgroups, respectively, for extensive lipidome analysis. Although this number of patients may seem low (n = 10, 5 patients in each group), to our knowledge, this type of characterization of the lipidome in samples from AML patients has never before been published. In addition, the number of independent measurements for this type of assay is classically n = 3m and in our case, we had five independent values for each parameter; this gives statistical significance to our results, which are quite robust.

We found an increase in the quantity of neutral lipids (FFAs), which are storage lipids, in patients with an unfavorable prognosis, suggesting that the mobilization of these storage lipids improve the viability of leukemic cells. Moreover, the increase in the quantity of phospholipids, and in particular, of sphingomyelin and phosphocholine (PC), in patients with an unfavorable prognosis confirms that PLs are markers of aggressiveness and tumor progression, which are associated with a poor prognosis in AML. These results are consistent with those described in cancers [26,27]. Elevated levels of PC indicate intense cell proliferation, commonly observed in cancers, in association with an increase in phosphoethanolamine (PE) [26]. At the neurological level, two studies have shown that patients with a low-grade glioma that progressed to a high-grade glioma had much higher levels of PC and PE [28,29]. Similarly, Wang et al. observed that PC was expressed at a higher level in the sera of patients with an intermediate prognosis compared to patients with a favorable prognosis. This is in line with an unfavorable impact of PC on prognosis [30]. The fact that we observed, in a previous study, that an increase in PC was a common mechanism of survival against starvation in leukemic cell lines [18] strongly suggests that its prognostic role in patients could be explained by a “protective” role of PC, which could increase the resistance of leukemic cells in stressful situations. It is very interesting to note that in this study, we observed a decrease in phosphatidyl-choline in a situation of metabolic stress in our cell lines. Taking into account that the catabolism of phosphatidyl-choline can produce phosphocholine, we can hypothesize that leukemic cells use phosphatidyl-choline catabolism to generate lipid mediators that will increase their survival. Moreover, a team recently created an inventory of lipidome in the plasma of patients with AML, and showed a decrease in total FFAs, including PC; this was probably be due to increased oxidation of FAs in AML cells. Altogether, these data underline the crucial role of phosphatidyl-choline and phosphocholine metabolism in the survival of leukemic cells, suggesting that it could be also an interesting therapeutic target.

In conclusion, our data show that balanced lipid metabolism is a phenotypic marker of the diversity (very probably genetic) of leukemic cells that significantly influences their proliferation and resistance to stress, and could be a fruitful therapeutic target to improve the survival of some AML patients harboring a particular “lipidic phenotype”.

## 4. Materials and Methods

### 4.1. Cell Culture and Conditions

The cells used for this experiment were purchased from the ATCC (American Type Culture Collection, Manassas, VA, USA) society. Five human leukemic cell lines were used in our study: HL60, HEL, OCI-AML3, KG1 and K562. The K562 line was derived from a patient who initially had chronic myeloid leukemia, but the cells were collected from this patient in the phase of transformation into AML. It is therefore considered a model of AML. The cells were cultured at a density of 0.5 × 10^5^ cells/mL for exponential growth in a medium containing 10% fetal bovine serum (FBS), 1% L-glutamine, 1% penicillin/streptomycin and RPMI 1640. The cells were incubated in a laboratory oven at 37% and 5% CO_2_, and the media were changed every two–three days.

The conditions used for the experiments were a control, a culture with etomoxir, starvation (medium without serum) and a culture with etomoxir under the conditions of starvation. Etomoxir sodium salt hydrate was purchased from Sigma Aldrich (Saint-Quentin-Fallavier, France) and reconstituted in sterile water for injection into a stock solution of 10 mM. For the cultures with etomoxir, the cells were suspended in a routinely used medium, and etomoxir was diluted directly with the medium to the desired concentration (100 µM, 200 µM or 400 µM). For starvation, the cells were centrifuged and resuspended in a medium deprived of FBS, with 1% L-glutamine, 1% penicillin/streptomycin, RPMI 1640 and 1% insulin–transferrin–selenium. The cultures with etomoxir under the conditions of starvation were prepared using the same protocol as the cultures with etomoxir under non-starvation conditions.

### 4.2. Fluorescence-Activated Cell Sorting

An amount of 0.3 × 10^6^ cells were centrifuged at 1400 rpm for 10 min and washed twice with phosphate-buffered saline (PBS) solution. The cells were labeled with 5 µL of Annexin V (FITC Annexin V Apoptosis Detection Kit I from BD Pharmingen, France) and 5 µL of Propidium iodide (PI), and incubated for 15 min at room temperature, protected from the light. Their acquisition was conducted using a FACS Lyric cytometer. Data analysis was performed using BD FACSuite Software. Handling was performed in parallel for each cell line and in three independent experiments.

### 4.3. Mitochondrial Respiration Assay

The bioenergetics assay was performed using the Seahorse XF Cell Mito stress test kit and the Seahorse XFe96 analyzer, purchased from Agilent Technologies. The day before the assay, the tissue culture plate was coated using Corning Cell-Tak adhesive, according to the supplier’s recommendations. The Seahorse cartridge was hydrated with sterile water and incubated overnight in a CO_2_-free incubator, at 37 °C. The day of the assay, the Seahorse medium was prepared, and the pH solution adjusted to 7.4. The following step involved seeding the cells on the previously coated culture plate. The cells were enumerated and centrifuged for 5 min at 2000 rpm, and the supernatant was removed and replaced with Seahorse medium. The cells were then incubated for 45 min in a CO_2_-free incubator, at 37 °C. During the incubation, the stock solutions were reconstituted and diluted to the appropriate concentrations as follows: ATP synthase inhibitor oligomycin at 1.5 µmol/L, uncoupling agent carbonyl cyanide p-trifluoro-methoxyphenyl hydrazone (FCCP) at 0.5 µmol/L and 2 µmol/L, fatty acid oxidation inhibitor etomoxir at 4 µmol/L, and inhibitors of complex I/III of the electron transport chain antimycin/rotenone at 0.5 µmol/L. The oxygen consumption rate (OCR) was measured at baseline and after sequential injections with the previously mentioned solutions.

### 4.4. Determination of IC_50_

The half-maximal inhibitory concentration (IC_50_) was determined using the PrestoBlue cell viability reagent, purchased from Thermo Fisher Scientific (Waltham, MA, USA). The cell line K562 was cultured at an initial density of 40,000 cells/well with increasing concentrations of etomoxir (0, 50, 100, 150, 200, 250, 300 µM), for 24 h, on a black culture plate with a flat, clear bottom. After 24 h, 10 µL of PrestoBlue reagent was added to each well. The cells were incubated for 30 min at 37 °C. The fluorescence was measured using a Varioskan microplate reader (Thermo Fisher Scientific).

The measurements were performed in triplicate for each condition. The average fluorescence was calculated for the triplicates corresponding to each concentration, and the background (blank) was subtracted.

### 4.5. Lipid Analysis

The cells were collected after 24 h of treatment, followed by quenching in an ethanol/dry ice bath, to stop cell metabolism. After quenching, the pellets were washed twice with PBS and stored at −80 °C until the day of the lipid extraction. Total lipids were extracted in chloroform/methanol (1:2) with an internal standard (10 nmol tridecanoic acid C13:0 and 10 nmol PC C23:0). By adding chloroform and 0.2% KCl/0.5% acetic acid, biphasic separation, at a chloroform/methanol/water ratio of 2:1:0.8 (*v/v/v*), was generated. The lowest organic phase that was recovered was the total lipid fraction. Total lipid equivalent of 0.2–0.5 M of the cells were dried in vacuo, and the derivatizing agent TMSH (Machery-Nagel, Hoerdt, France) in chloroform/methanol (1:1, *v/v*) was added, followed by gas chromatography–mass spectrometry (GC-MS, Agilent 5977A-7890B). The total fatty acid amount was quantified using Mass Hunter software (Agilent, Santa Clara, CA, USA) in the form of fatty acid methyl esters, using the calibration curve generated using the authentic standards, followed by normalization using the internal standards and cell number.

### 4.6. Plasma Analysis

We recovered the plasma of 39 adult patients with a diagnosis of de novo AML, who received an induction treatment according to the standards for “fit” patients. All subjects gave their informed consent for inclusion before they participated in the study. The study was conducted in accordance with the Declaration of Helsinki, and the protocol was approved by the Ethics Committee of CHUGA (Project 38RC13.209). The patients’ prognoses were assessed according to the ELN 2017 classification: 21 patients belonged to the favorable prognosis group and 18 to the unfavorable prognosis group. Plasma was collected upon diagnosis of AML. All the characteristics of the patients are summarized in Appendix A. The analysis of total FAs and lipid species was carried out according to the protocol described in the section “Lipid Analysis”. Detailed information on the composition of the lipid species was determined for 5 patients representative of each of the two prognostic subgroups. Thereby, we were able to identify 8 compounds in these 10 patients. Sphingomyelin (SM), phosphatidyl-choline ((P)C), phosphatidyl-inositol ((P)I) and phosphatidyl-ethanolamine ((P)E), which correspond to phospholipids (PLs), appeared after the first migration. On the other hand, a mixture of diacylglycerol and cholesterol (DAG-Cho), free fatty acids (FFAs), triacylglycerol (TAG) and cholesteryl esters (CEs), which correspond to neutral lipids (NLs), appeared after the second migration.

### 4.7. Statistical Analysis

For statistical analysis, the Kruskal–Wallis test was used for three or more groups, and the Mann–Whitney test was used to compare two groups. Unless otherwise specified, the experiments were repeated at least three times. The data were analyzed using GraphPad Prism software (version 7). A *p*-value < 0.05 was considered significant.

## Figures and Tables

**Figure 1 ijms-24-05988-f001:**
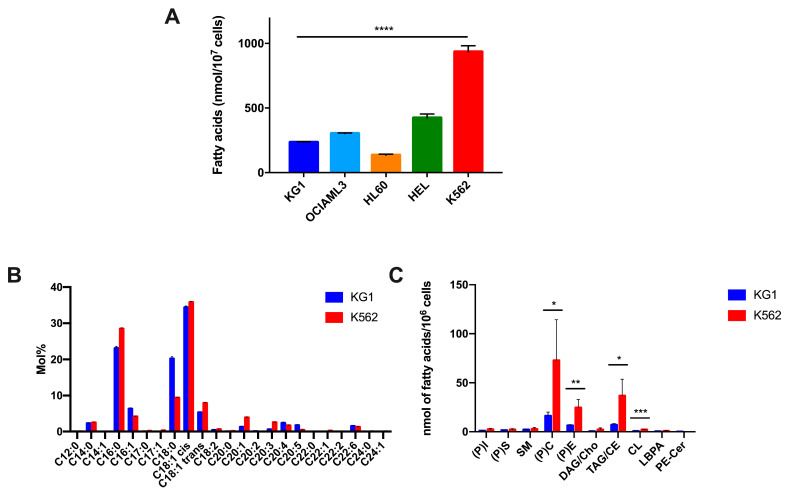
Characterization of the lipid profiles of the leukemic cell lines at basal state. (**A**) Total fatty acid quantification in the five leukemic cell lines. (**B**) Composition of total fatty acids in KG1 and K562 cells. (**C**) Lipid species quantification in KG1 and K562 cells. * *p* < 0.05, ** *p* < 0.01, *** *p* < 0.001, **** *p* < 0.0001. Abbreviations: (P)I: phosphatidyl-inositol, (P)S: phosphatidyl-serine, SM: sphingomyelin, (P)C: phosphatidyl-choline, (P)E: phosphatidyl-ethanolamine, DAG: diacylglycerol, Cho: cholesterol, TAG: triacylglycerol, CE: cholesteryl-ester, CL: cardiolipin, LBPA: lyso-biphosphatidic acid, Cer: ceramide.

**Figure 2 ijms-24-05988-f002:**
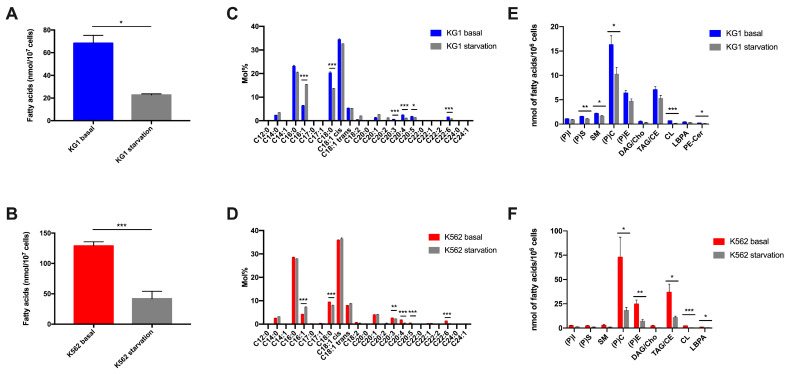
Impact of the nutrient depletion on the two cell lines KG1 and K562. (**A**,**B**) Total fatty acid amount in KG1 and K562, respectively, at basal state versus in starvation. (**C**,**D**) Composition of total fatty acids in KG1 and K562, respectively, at basal state versus in starvation. (**E**,**F**) Lipid species quantification in KG1 and K562, respectively, at basal state versus in starvation. * *p* < 0.05, ** *p* < 0.01, *** *p* < 0.001.

**Figure 3 ijms-24-05988-f003:**
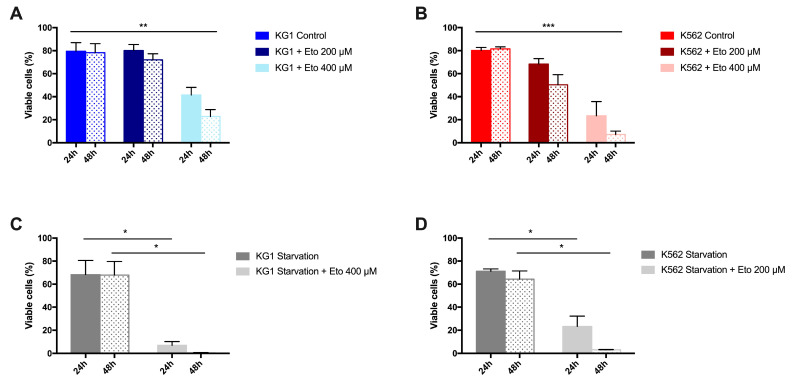
Impact of etomoxir and/or starvation on cell proliferation in KG1 and K562. (**A**,**B**) Percentage of viable cells in basal state, and with two concentrations of etomoxir (200 and 400 µM) in KG1 and K562 cells, respectively. (**C**,**D**) Percentage of viable cells in starvation and in starvation with etomoxir in KG1 and K562 cells, respectively. * *p* < 0.05, ** *p* < 0.01, *** *p* < 0.001.

**Figure 4 ijms-24-05988-f004:**
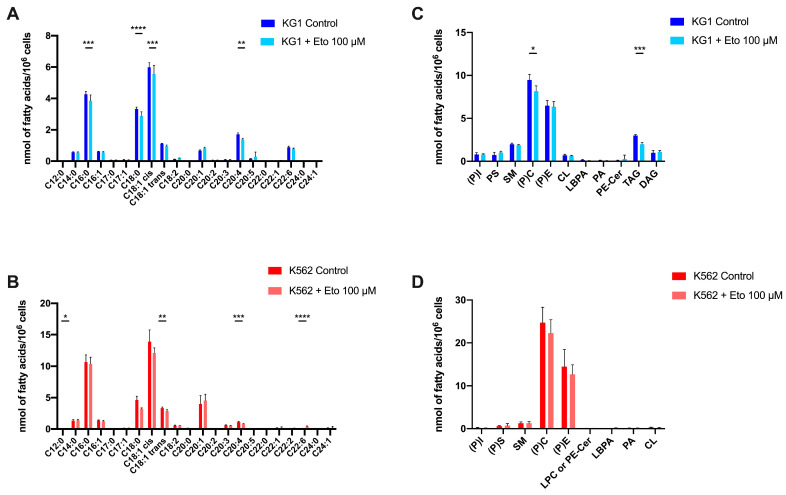
Impact of a low dose of etomoxir on the two cell lines KG1 and K562. (**A**,**B**) Composition of total fatty acids in KG1 and K562, respectively, at basal state versus with 100 µM etomoxir. (**C**,**D**) Lipid species quantification in KG1 and K562, respectively, at basal state versus with 100 µM etomoxir. * *p* < 0.05, ** *p* < 0.01, *** *p* < 0.001, **** *p* < 0.0001.

**Figure 5 ijms-24-05988-f005:**
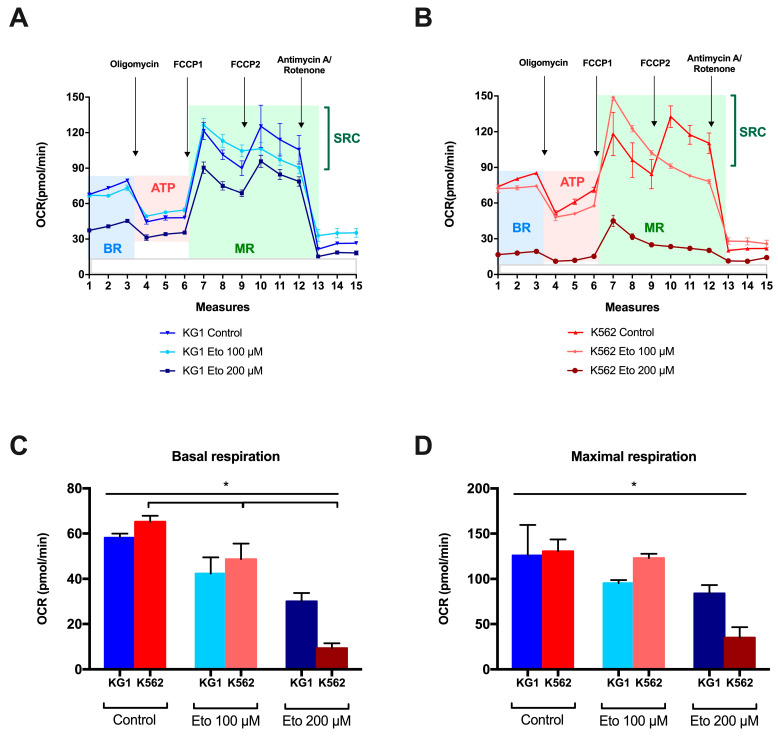
Variation in the respiratory profiles of the two cell lines KG1 and K562 at 24 h of culture. The profiles of the key parameters of KG1 (**A**) and K562 (**B**) in their basal state, and with increasing concentrations of etomoxir (100 and 200 µM) are presented, as are the OCR variations in basal respiration (**C**) and maximal respiration (**D**). * *p* < 0.05. Abbreviations: ATP: adenosine triphosphate, BR: basal respiration, FCCP: carbonyl cyanide p-trifluoro-methoxyphenyl hydrazone, MR: maximal respiration, OCR: oxygen consumption rate, SRC: spare respiratory capacity.

**Figure 6 ijms-24-05988-f006:**
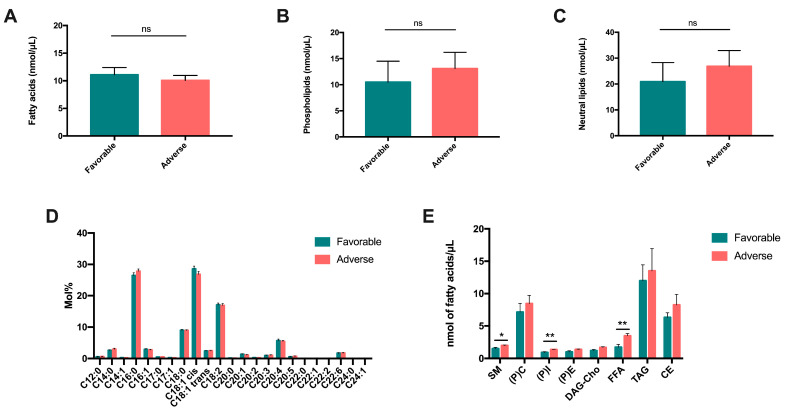
Variation in fatty acid content and lipid species in AML patients according to their prognosis. The amounts of total fatty acids (**A**), phospholipids (**B**) and neutral lipids (**C**) were quantified in patients with favorable and adverse prognoses. (**D**) Composition of total fatty acids in plasma of patients with favorable and adverse prognoses. (**E**) Lipid species quantification in plasma of patients with favorable (n = 5) versus adverse (n = 5) prognoses. ns: non-significant, * *p* < 0.05, ** *p* < 0.01.

## Data Availability

Raw data supporting reported results are available by asking to the corresponding authors.

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
