# Peer review of "Variation in Lipid Species Profiles among Leukemic Cells Significantly Impacts Their Sensitivity to the Drug Targeting of Lipid Metabolism and the Prognosis of AML Patients"

_ijms, 2023, doi:10.3390/ijms24065988_

Round 1
Reviewer 1 Report
In the present study, Lo Presti et al. characterized the lipidomic profiles of AML cell lines and in plasma from AML patients. Results indicate differences in the lipid profiles at steady state, and suggest remodeling of lipid homeostasis in response to nutrient stress. Interestingly, sensitivity to Etomoxir, which blocks FAO, was dependent of the initial lipid profile of the cell lines. Furthermore, the lipid profiles of plasma from AML patients were significantly correlated with the patients’ prognosis.
This is an interesting observational study. The following minor changes would improve the quality of the manuscript.
· Line 24: FAO should be spelled out the first time.
· Lines 28-31: this sentence is rather convoluted and should be simplified.
· Line 111: change lipide to lipid.
· Lines 114-116: there is a paragraph 2.1.1, but there is no paragraph 2.1.2; the title on paragraph 2.1.1 seems redundant
· Lines 135-136: is this the title of a paragraph?
· Lines 165-165: the effects of serum starvation may be due to many other factors, e.g. depletion of growth factors as well as other molecular moieties bound to albumin or serum lipoprotein. This experiment is rather crude and the results should be validated in experiments that employ growth medium containing dialyzed serum and/or by using serum-free medium supplemented with short-chain FA (e.g. octanoate).
· Line 171: is this a new paragraph? The title seems incomplete.
· In general, the quality of the figures is poor. The figures should not include titles (to avoid redundancy with titles in the figure legends). The fonts are hard to read and are of mixed sizes (e.g. Fig. 3).
· Lines 284-288: the concentrations of Etomoxir are indicated as milli molar, while in the text are indicated as micro molar
· Paragraph 2.6: the number of patients analyzed appears to be quite low, especially considering the high clinical and genetic heterogeneity of AML.
Author Response
Reviewer-1
In the present study, Lo Presti et al. characterized the lipidomic profiles of AML cell lines and in plasma from AML patients. Results indicate differences in the lipid profiles at steady state, and suggest remodeling of lipid homeostasis in response to nutrient stress. Interestingly, sensitivity to Etomoxir, which blocks FAO, was dependent of the initial lipid profile of the cell lines. Furthermore, the lipid profiles of plasma from AML patients were significantly correlated with the patients’ prognosis
This is an interesting observational study. The following minor changes would improve the quality of the manuscript.
- Line 24: FAO should be spelled out the first time.
Lines 28-31: this sentence is rather convoluted and should be simplified.
- Line 111: change lipide to lipid.
- Lines 114-116: there is a paragraph 2.1.1, but there is no paragraph 2.1.2; the title on paragraph 2.1.1 seems redundant
- Lines 135-136: is this the title of a paragraph?
All these modifications have been done as requested
- Lines 165-165: the effects of serum starvation may be due to many other factors, e.g. depletion of growth factors as well as other molecular moieties bound to albumin or serum lipoprotein. This experiment is rather crude and the results should be validated in experiments that employ growth medium containing dialyzed serum and/or by using serum-free medium supplemented with short-chain FA (e.g. octanoate).
The reviewer's remark concerning the multi-factorial nature of serum deficiency is quite correct and does not allow us to conclude on the specific impact of a particular serum component. However, in this experiment, our goal was to study the adaptation of leukemic cells to a global metabolic stress and not to a particular deficiency. So, a global serum deprivation was adapted to the goal of this experiment. It’s a methodology that we have already used and published in our team (Lo Presti et al, Metabolomics 2020) which makes it possible to identify the major survival mechanisms of leukemic cells. The use of serum free medium selectively supplemented with a particular type of lipid species, as proposed by the reviewer, would be more precise but would not have allowed us to have an overall picture of lipid reprogramming in this situation, which was our objective.
Moreover, the FBS is the major source of lipids in the medium and the removal of serum is now valid and used in a number of publications (Dass et al. Nature Com. 2021...). In addition, to date there is no controlled medium whose exact and necessary FA composition is known, so an FBS-free medium supplemented only with C18:0 or C18:1 would also have posed problems of interpretation.
- Line 171: is this a new paragraph? The title seems incomplete.
We corrected the title to avoid mis-understanding.
- In general, the quality of the figures is poor. The figures should not include titles (to avoid redundancy with titles in the figure legends). The fonts are hard to read and are of mixed sizes (e.g. Fig. 3).
We agree with the reviewer that the quality of some figures is low and not admissible for quality publication. All the figures have been properly redone and harmonized according to the recommendations of the reviewers. Thereby, in the new version of the manuscript the quality and clarity of the figures have been significantly improved. ·
- Lines 284-288: the concentrations of Etomoxir are indicated as milli molar, while in the text are indicated as micro molar
We corrected it
- Paragraph 2.6: the number of patients analyzed appears to be quite low, especially considering the high clinical and genetic heterogeneity of AML.
The number of patients studied for this experiment may indeed seem low (n=10, 5 patients in each group). It should be noted that first of all our initial cohort included 39 AML patients at diagnosis. In addition, the investigation of the lipid profiles of all lipid species (FA and phospholipid composition) by LC-MS and GC-MS is a very time-consuming experience that requires very high-level expertise. Therefore, these are not investigations that can be carried out on large series of patients. In fact, to my knowledge, this type of characterization of the lipidome on samples from AML patients had never before been published. In addition, the number of independent measurements for this type of assay is classically n=3 and in our case, we have 5 independent values for each parameter which gives a statistical significance of our results which are quite robust.
Reviewer 2 Report
The manuscript describes the lipidomic profiles of AML cell lines under steady state conditions or under the pressure of FBS deprivation. The same conditions are also used to test sensitivity to the FAO targeting agent Etomoxir. Moreover, the author analyze the plasma lipidomic content of primary AML samples and show some differences related to AML risk classification. The topic of the manuscript is of potential interest, however some major points lower its merits. The analyzed AML cell lines are representative of few AML subtypes and in particular the authors choose K562 cells, that are known as a chronic myeloid leukemia model, for downstream assays. Moreover, the analysis of plasma from primary patients is not linked to the rest of the manuscript.
Author Response
Reviewer2
The manuscript describes the lipidomic profiles of AML cell lines under steady state conditions or under the pressure of FBS deprivation. The same conditions are also used to test sensitivity to the FAO targeting agent Etomoxir. Moreover, the author analyze the plasma lipidomic content of primary AML samples and show some differences related to AML risk classification. The topic of the manuscript is of potential interest; however, some major points lower its merits.
- The analyzed AML cell lines are representative of few AML subtypes and in particular the authors choose K562 cells, that are known as a chronic myeloid leukemia model, for downstream assays.
First of all, we would like to thank the reviewer for his encouraging opinion on our manuscript. We fully agree on the limits of human leukemic cell lines as a model for AML investigations. It is for this reason that we first evaluated 5 different leukemic cell lines which were representative of a certain diversity of AML, even if of course they do not represent all the subtypes of AML.
K562 is indeed a line derived from a patient initially in CML but the cells were collected in the transformation phase into AML of this patient. It is therefore considered as a model of AML. Moreover, our choice of K562 and KG1 was not based on the genotype and/or phenotypic characteristics of these cell lines but because the 2 cell lines presented very different levels of FA at baseline, which suggested very different lipids composition. In fact, the qualitative analysis of their lipid contents showed very clear differences, demonstrating the diversity of the basic lipidome of leukemic cells.
- Moreover, the analysis of plasma from primary patients is not linked to the rest of the manuscript.
Regarding the analysis of patient plasma, as underlined by the reviewer, leukemic cell lines represent a questionable model of what happens in patients. This is why it seemed to us informative and instructive to reinforce the in vitro analysis with an ex-vivo analysis on patient samples, in order to assess the impact of the patients' lipidome on their survival. In fact, we believe that the results highlighted on these patients provide extremely interesting elements about the impact of the expression of certain phospholipids on the prognosis of patients
Reviewer 3 Report
Some points regarding the manuscript can be stated:
Almost all references are inappropriate or wrong.
The percentage of viable cells related to each cell line is not consistent in different plots.
The figure legends are not complete and the signs are not explained everywhere, also there are variations in the abbreviations, for example, PC is used for both Phosphocholine and Phosphatidylcholines.
Some figures are of very low quality.
Some claims don't seem right, for example, CPT1 is introduced as responsible for importing FA, while carnitine–acylcarnitine translocase does it and...
Author Response
Reviewer3
Comments and Suggestions for Authors
Some points regarding the manuscript can be stated:
- Almost all references are inappropriate or wrong:
Firstly, we want to apologize for the very low quality of the editing of our references in the submitted manuscript. In the new version of the revised manuscript all the references were checked and reintroduced in the manuscript, and the table was re-edited properly. We hope that now the bibliography is appropriate and right.
- The percentage of viable cells related to each cell line is not consistent in different plots.
- The figure legends are not complete and the signs are not explained everywhere, also there are variations in the abbreviations, for example, PC is used for both Phosphocholine and Phosphatidylcholines.
- Some figures are of very low quality.
We provide a global answer to the 3 previous remarks.
Concerning the figure, all the figures have been properly redone. In particular we corrected and completed the legends and we harmonized the abbreviations and the expression of results. Moreover, we have improved the quality of the figures by re-editing some of them.
We hope that now the figures and their legends are clear and will allow the reader to clearly understand our results.
- Some claims don't seem right, for example, CPT1 is introduced as responsible for importing FA, while carnitine–acylcarnitine translocase does it
In fact, CPT1A is the abbreviation for the Carnitine Palmitoyl-carnitine Translocase 1, which is responsible for fatty acid transfer in mitochondria. It is the canonical transporter acyl-carnitine translocase in all the eukaryotes cells.
- There are other cases that appear to be more than a typo, for example FCCP is introduced as "carbonyl cyanide p-trifluoro-methoxyphenyl hydrazine" when this abbreviation is equivalent to "carbonyl cyanide p-trifluoro-methoxyphenyl hydrazone". Hydrazone and hydrazine are different compounds.
Thanks to the reviewer to have identified this error. We corrected it
- Apart from these, there are typographical errors in some places for example line 69, 233, and 296.
Overall, the manuscript has been carefully proofread to correct all typographical errors. We also had it proofread by a person of English nationality to check the quality of the English and the absence of nonsense or false meaning.
Reviewer 4 Report
Summary: In the present study, the authors performed mass spectrometry-based lipidomics analyses on AML and CML cell lines cultured in the presence or absence of the CPT1a inhibitor, etoxomir, under normal versus starvation conditions. The data are novel and interesting; however, the presentation of the data needs improvement. Specific comments to improve the manuscript are included below.
1) Abstract Lines 29-31: “In conclusion, our data showed that balance between lipid species is a phenotypic marker of the diversity of leukemic cells, witness of lipid metabolism reprogramming. That significantly influence their proliferation and resistance to stress, and the prognosis of AML patients.”
a. What do you mean by ‘witness of lipid metabolism reprogramming’? Please clarify.
b. Should this be one sentence instead of two? The second sentence does not make sense as written. Please revise accordingly.
2) Introduction Lines 100-104: In the last paragraph of the Introduction, please include a clearly stated hypothesis that summarizes the rationale for performing the study.
3) A recent study demonstrated that glycerophospholipid metabolism is involved in drug resistance of chronic myeloid leukemia. Since K562 cells were included in this study, please reference the paper (PMID 36536477) and discuss either in the Introduction or Discussion.
4) Figure 1: Are error bars included on all bars of the graphs? Some are visible whereas others are not. They would be easier to see if they were black instead of the same color as the bars. Is Figure 1B a repeat of the data presented in Figure 1A? They look identical. If so, that panel can be removed.
5) Line 177: Etoxomir is a CPT1a inhibitor. Please include this information somewhere in the introduction.
6) Figure 3: Is panel C a repeat of the data presented in panels A-B? If so, that panel can be removed. Similarly, is the control data in panels D-E a repeat of the data presented in panels A-B? They look identical. Please only report data values once in the manuscript.
7) Figure 4: Can the authors explain why a CPT1a inhibitor resulted in a decrease of lipid species in KG1 and K562 cells?
8) Figure 5A: Please split the KG1 and K562 data onto two separate OCR graphs so it is easier to compare the treatment groups.
9) Figure 5D-E: Are there statistics to provide for the data in these panels?
10) The legend for Figure 5E should also define the other abbreviations (BR, ATP, MR, SRC, FCCP1, FCCP2).
11) Please provide a table summarizing the patient characteristics for favorable (n=21) versus unfavorable (n=18) AML patients used in this study.
12) The Discussion is very much just a repeat of the Results section. Please revise to include more about the overall implications of your findings to the field.
13) Materials and Methods Line 417: The cells were incubated in an autoclave? I hope not! Please revise.
Author Response
Dear reviewer
We provide in the attached file ' Responses to reviewer 4" a complete and “point by point” answer to your remarks.
We hope that these modifications and additions have improved our manuscript and answered to all your interrogations.
We want to stress that, following previous recommendations and comments of reviewer 1 and 2, we have also integrated in the manuscript some elements of the responses that we have provided to these two reviewers. These elements are highlighted in yellow in the text to differentiate them to modifications done following your comments (highlighted in blue). We hope that these “marks” will allow each reviewer to clearly see modifications of the text that have been done according to their recommendations.
We hope that these modifications fulfill your expectations and that now our manuscript will be acceptable for publication
Best feeling

Round 2
Reviewer 2 Report
The limitations of the study have not been addressed in the revised version.
Author Response
Dears reveiwers
Following your last recommendation we have integrated in the manuscript the different comments that we provided to your first revision. These comments are highlighted in yellow in the text, so you can check easily. I also inform you that following comments and remarks of reviewer 4 we made some additional modifications in the text ( mainly in the discussion). These modifications are highlighted in blue to avoid confusion .
I hope that now our manuscript is more clear.
best feeling
Reviewer 4 Report
The authors have adequately addressed my concerns.